# Beyond Binary: Towards Fine-Grained LLM-Generated Text Detection via Role Recognition and Involvement Measurement

## Abstract

The rapid development of large language models (LLMs), like Chat-GPT, has resulted in the widespread presence of LLM-generated content on social media platforms, raising concerns about misinformation, data biases, and privacy violations, which can undermine trust in online discourse. While detecting LLM-generated content is crucial for mitigating these risks, current methods often focus on binary classification, failing to address the complexities of real-world scenarios like human-AI collaboration. To move beyond binary classification and address these challenges, we propose a new paradigm for detecting LLM-generated content. This approach introduces two novel tasks: LLM Role Recognition (LLM-RR), a multi-class classification task that identifies specific roles of LLM in content generation, and LLM Influence Measurement (LLM-IM), a regression task that quantifies the extent of LLM involvement in content creation. To support these tasks, we propose LLMDetect, a benchmark designed to evaluate detectors' performance on these new tasks. LLMDetect includes the Hybrid News Detection Corpus (HNDC) for training detectors, as well as DetectEval, a comprehensive evaluation suite that considers five distinct cross-context variations and multi-intensity variations within the same LLM role. This allows for a thorough assessment of detectors' generalization and robustness across diverse contexts. Our empirical validation of 10 baseline detection methods demonstrates that fine-tuned Pre-trained Language Model (PLM)-based models consistently outperform others on both tasks, while advanced LLMs face challenges in accurately detecting their own generated content. Our experimental results and analysis offer insights for developing more effective detection models for LLM-generated content. This research enhances the understanding of LLM-generated content and establishes a foundation for more nuanced detection methodologies.

## CCS Concepts

• **Do Not Use This Code** → **Generate the Correct Terms for Your Paper**; *Generate the Correct Terms for Your Paper*; Generate the Correct Terms for Your Paper; Generate the Correct Terms for Your Paper.

## Keywords

Social Media, Large Language Models, LLM-generated Text Detection, AI-assisted News Detection

**ACM Reference Format:**

Anonymous Author(s). 2018. Beyond Binary: Towards Fine-Grained LLM-Generated Text Detection via Role Recognition and Involvement Measurement. In *Proceedings of Make sure to enter the correct conference title from your rights confirmation emai (Conference acronym 'XX).* ACM, New York, NY, USA, 13 pages. https://doi.org/XXXXXXX.XXXXXXX

## 1 INTRODUCTION

> "On the internet, nobody knows you're a ~~dog~~ AI."
>
> — *Peter Steiner*

Recent advances in generative large language models (LLMs) [15, 16, 25, 35], such as GPT-4 [35] and LLaMA [16], alongside the increasing availability of tools like ChatGPT[1] and Copilot[2], have significantly reshaped the landscape of social media and web platforms [13]. These technologies facilitate the automated creation of extensive content with human-like fluency [36], making LLM-generated posts, articles, and comments widely accessible and rapidly disseminated. The proliferation of such content has profoundly expanded its reach and influence, transforming the dynamics of online discourse.

However, these advancements also introduce significant risks, both in terms of information accuracy and public trust. While LLM-generated content can match the fluency of professional writing, it inevitably contains hallucinations [1, 4]—misleading information that appears credible but lacks factual accuracy. A report by NewsGuard[3] identified over 1,050 unreliable LLM-generated news websites, further undermining the already fragile information ecosystem. The rapid spread of such content across social media heightens the risk of misinformation [37, 40], challenging the accuracy and credibility of digital information. Additionally, LLM-generated content often exhibits inherent biases [18] and can be misused for malicious purposes [33, 47], further complicating efforts to maintain information integrity. These risks contribute to the erosion of public trust in media. According to the 2024 Digital News Report [5], global trust in news media has fallen to 40%, and the rise of LLM-generated content threatens to further weaken this fragile trust. As distinguishing between human-written and LLM-generated content becomes critical for preserving information integrity [11, 52], current detection methods, which are largely limited to binary classification [44, 46, 48, 49], fail to distinguish the complexity of LLM-generated content like mixed human-LLM input.

In real-world applications, LLMs play diverse roles, adapting to various user needs [9, 10]. These models assist in different stages of the writing process—from organizing ideas and drafting to refining text—resulting in varying degrees of AI involvement across contexts. Fully LLM-generated content is generally easier to detect

---

[1] https://chatgpt.com/

[2] https://copilot.microsoft.com/

[3] https://www.newsguardtech.com/special-reports/ai-tracking-center/

and often lacks quality assurance, whereas human-authored drafts refined by LLM tend to achieve higher quality standards and are significantly harder to identify as LLM-generated [53]. This variation complicates the detection process, underscoring the limitations of binary classification methods and the need for more advanced frameworks capable of capturing these nuanced distinctions.

In this paper, we propose a new paradigm for detecting LLM-generated content that moves beyond the limitations of binary classification by considering both the LLM's role and level of involvement in content creation, as depicted in Figure 1. LLMs often play diverse roles in assisting human authors, and to capture this complexity, we introduce two novel tasks. The first task, LLM Role Recognition (LLM-RR), is a multi-class classification task that identifies the specific roles played by LLMs in content generation, distinguishing between stages such as drafting and refinement. The second task, LLM Influence Measurement (LLM-IM), is a regression task designed to quantify the LLM involvement ratio in content creation, offering a nuanced measure of AI influence on the final output.

To evaluate these tasks, we present LLMDetect, a benchmark specifically designed to assess detection models' performance in real-world scenarios. LLMDetect consists of two components: the Hybrid News Detection Corpus (HNDC), a dataset with diverse content types for robust training and evaluation, and DetectEval, a comprehensive evaluation suite that considers five distinct cross-context variations and multi-intensity variations within the same LLM role. Together, these components provide a thorough assessment of detection model robustness and generalization across different contexts of LLM-generated content.

We validate our approach by training and evaluating 10 baseline detection models on the HNDC, including zero-shot LLMs, as well as supervised feature-based and Pre-trained Language Model (PLM)-based models. Our results show that fine-tuned PLM-based methods consistently outperform others in both tasks, while advanced LLMs face challenges in accurately detecting their own generated content. Specifically, DeBERTa-based detectors excel in cross-context generalization due to their advanced contextual representation capabilities, while Longformer-based models perform best on datasets with varying intensity levels, benefiting from their ability to process longer input sequences. Additionally, we investigate the impact of data leakage on zero-shot LLM detectors and explore the effect of using different LLMs as feature extractors. These findings demonstrate the effectiveness of our approach in handling complexities of LLM-generated content. Our **contributions** are summarized as follows[4]:

- We propose a new detection paradigm that moves beyond binary classification, introducing two novel tasks: LLM Role Recognition (LLM-RR) and LLM Influence Measurement (LLM-IM).
- We introduce LLMDetect, a benchmark comprising the Hybrid News Detection Corpus (HNDC) and DetectEval, designed to evaluate model robustness and generalization across diverse real-world content types.

- We empirically validate baseline detection methods, including zero-shot LLMs, supervised feature-based and PLM-based models. Our results offer insights for developing more effective detection models for LLM-generated content.

## 2 RELATED WORK

### 2.1 Detection tasks

As the growing number of LLMs continues to exhibit strong text generation capabilities [15, 16, 25, 35], several studies have begun to focus on the detection of AI-generated text. The early work mainly focused on distinguishing the pairs of human answers and GPT-generated answers for the same question, such as the HC3 dataset [21]. Subsequently, the work gradually shifted towards a broader range of scenarios. Some works expand the samples generated by a single LLM to various LLMs (such as MGTBench [23]) in multiple domains (essays, stories, and news articles). Moreover, since the writing style and language bring a significant challenge to the detector [30], other work [32, 49] focuses on detecting text generated by different LLMs in multiple languages. Recent works start from the generation method and focus on a broader range of AI-assisted writing methods, such as from GPT-generated to GPT polished [53], as well as GPT-completed [31, 43].

However, existing work usually focused on binary classification tasks [51], which determine whether it is human-written or not, ignoring the differences in which humans integrate ChatGPT into their creations in real-life scenarios, such as complete generation, continuation, and polishing [31, 43]. In contrast, our work introduces a more nuanced detection framework, addressing these complexities by accounting for different levels of LLM involvement, offering a more detailed and practical understanding of LLM-generated content.

### 2.2 Detection methods

Current detection methods can be broadly categorized into three types based on the features they rely on [20]: watermarking-based detection methods, statistical outlier detection methods, and fine-tuning classifiers. (i) The watermark-based methods require embedding the signals that are invisible to humans into the AI-generated text and then detecting them based on these invisible token-level secret markers [26]. However, this method not only requires pre-editing that is not applicable to open-source models [37] but also affects the quality of model generation due to the insertion of watermarks [42]. (ii) Statistical outlier detection methods focus on distinguishing whether a text is written by GPT based on the human features contained in the text. They adopt features ranging from shallows (entropy [19, 28], n-gram frequencies [2], and perplexity [7]) to deeps such as using the absolute rank [19], the Log Likelihood Ratio Ranking (LRR) by complementing Log Rank [41] and the model's log probability in regions of negative curvature (DetectGPT) [34]. (iii) Supervised fine-tuning classifiers, trained on annotated data [3, 24, 39], have shown effectiveness in detecting LLM-generated text across domains, such as news [27, 54], social media (e.g., Twitter) [17], and academic papers [53]. However, these classifiers often overfit to specific domains, leading to poor performance on out-of-distribution data [12, 45], and their capabilities

---

[4]Our benchmark and trained detection models will be released.

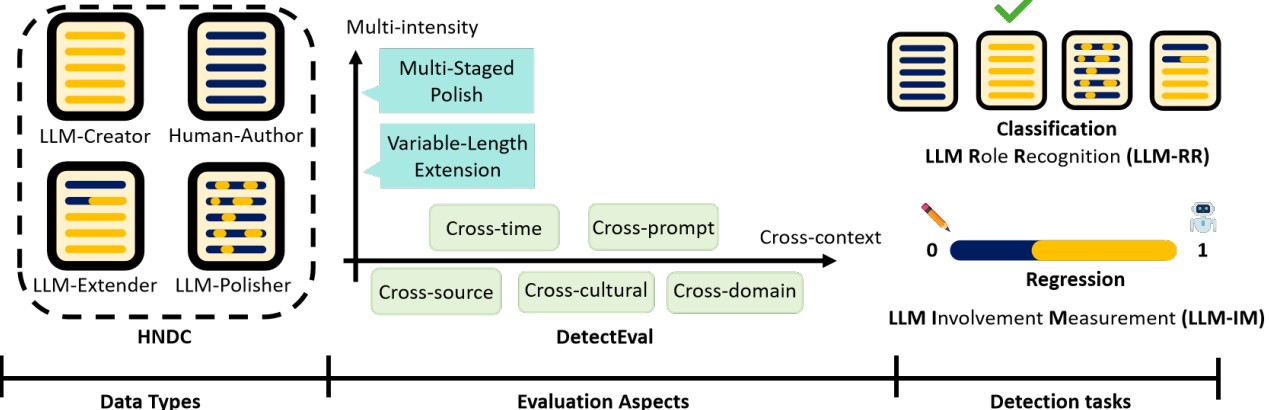

**Figure 1: The detection framework toward fine-grained LLM-generated text detection through role recognition and involvement evaluation.**

significantly degrade when applied to unseen datasets from different domains [29].

Thus, assessing the transferability of detection models is essential for their practical application across diverse datasets and domains. In our work, we evaluate detection methods—including zero-shot LLMs, supervised feature-based, and PLM-based models—on our novel tasks and dataset. Our empirical validation provides insights into improving detection models for LLM-generated content, particularly with respect to enhancing generalization and robustness in diverse, real-world scenarios.

## 3 METHODOLOGY

Figure 1 illustrates our proposed detection framework for fine-grained detection of LLM-generated text, encompassing two proposed novel detection tasks (§3.1), and the LLMDect Benchmark (HNDC & DetectEval) for training and evaluation detection models (§3.2).

### 3.1 Detection Paradigm Definition

The current task of detecting LLM-generated text primarily relies on binary classification, determining whether a text is LLM-generated or not. In this traditional binary LLM detection task, given a dataset $\{(x_i, y_i)\}_{i=1}^N$, where $x_i$ denotes the text content and $y_i \in \{0, 1\}$ indicates whether the text is LLM-generated. However, this approach focuses only on identifying LLM-generated content and is unable to distinguish more complex scenarios. For example, in LLM-assisted writing, users may employ LLMs to refine or slightly modify sentence structures for improved fluency, which is different from cases where the LLM generates the entire text. To overcome these limitations, we propose two new detection tasks: **LLM Role Recognition (LLM-RR)**, a multi-class classification task, and **LLM Involvement Measurement (LLM-IM)**, a regression task.

*3.1.1 LLM Role Recognition.* LLM-RR aims to identify the specific role that LLM plays in text generation when used as an LLM-assisted writing tools. Unlike binary detection, where labels are binary, the label for each $x_i$ is defined as $y_i \in \{C_1, C_2, \ldots, C_k\}$, indicating the specific role $C_i$ the LLM plays in generating text $x_i$. Examples of such roles include fully human-written text, LLM-generated content with minor human editing, human-led creation with LLM assistance, or fully LLM-generated text without human involvement, among others. The LLM-RR task can be defined as follows:

$$\min_f \mathbb{E}_{(x,y)\sim\mathcal{D}} \left[ \mathbf{1}\{f(x) \neq y\} \right] \quad (1)$$

where our objective is to find an optimal classifier $f(\cdot)$ that minimizes the overall average misclassification rate. The indicator function $\mathbf{1}\{f(x_i) \neq y_i\}$ equals 1 when the classifier $f$ assigns an incorrect label to the input $x_i$, and 0 if the label is correct. The text $x_i$ follows a distinct distribution $\mathcal{F}_k$ conditioned on its category label $y_i$:

$$x_i \mid y_i = C_j \sim \mathcal{F}_j, \quad j \in \{1, 2, 3, \ldots, k\} \quad (2)$$

*3.1.2 LLM Involvement Measurement.* While LLM-RR provides greater granularity compared to binary detection, it also has limitations. First, user interactions with LLMs in real-world applications are complex, making it challenging to accurately define all possible LLM roles. Additionally, even within the same LLM role, the degree of the LLM's contribution can vary, adding further complexity to detection. Therefore, we propose a new task, LLM Involvement Measurement (LLM-IM), to address these challenges. LLM-IM quantifies the degree of LLM involvement in the generated text and is framed as a regression task. In this task, the dataset $\{(x_i, y_i)\}_{i=1}^N$ features labels $y_i$ that represent a continuous value rather than discrete role categories. We define this value label as the **LLM Involvement Ratio (LIR)**, a metric ranging from 0 to 1, where 0 indicates no LLM involvement, and 1 signifies that the text is entirely generated by the LLM. Specifically, it is calculated as follows:

$$LIR = \frac{T_{LLM}}{T_{total}} \quad (3)$$

where $T_{LLM}$ represents the portion of the text generated or edited by the LLM, $T_{total}$ represents the total length of the final text. The LLM-IM task can be defined as follows:

$$\min_f \mathbb{E}_{(x,y)\sim\mathcal{D}} \left[ (f(x) - y)^2 \right], \quad y \in [0,1] \tag{4}$$

where our objective is to minimize the expected loss over the data distribution $\mathcal{D}$.

By integrating the LLM-RR and LLM-IM tasks, this framework offers a comprehensive and scalable approach to understanding both the roles and the extent of LLM involvement in content creation.

## 3.2 LLMDect Benchmark

To validate the effectiveness of our proposed detection paradigm, we construct **LLMDect**, a benchmark specifically designed to evaluate detection models across varying levels of LLM involvement. This benchmark encompasses four distinct roles in content creation: Human-Author, LLM-Creator, LLM-Polisher, and LLM-Extender. Each of the four roles represents a distinct level of LLM participation:

- **Human-Author**: Content created entirely by a human, without any LLM intervention.
- **LLM-Creator**: Text fully generated by the LLM, with no human contribution.
- **LLM-Polisher**: Human-authored text that has been edited, refined, or improved by the LLM.
- **LLM-Extender**: Text where the LLM extends or continues an initial human-authored draft.

Specifically, the LIR is defined as 0 for Human-Author text and 1 for LLM-Creator text, while for LLM-Polisher and LLM-Extender text, the LIR falls between 0 and 1. For LLM-Extender, the LIR value can be directly calculated using Equation 3. However, for LLM-Polisher, directly extracting $T_{LLM}$ is not feasible. Therefore, following the approach of Yang et al. [53], we use the Jaccard distance to calculate the polish ratio, which serves as the LIR for this role.

In LLMDect, each Human-Author text is paired with three versions generated by the other LLM roles. Each text is annotated with its corresponding LLM role and associated LIR value. This dual annotation framework enables a comprehensive evaluation of both the role and extent of LLM involvement in content creation. The constructed LLMDetect benchmark comprises two key components: the **Hybrid News Detection Corpus (HNDC)**, a diverse dataset designed for robust training and evaluation of detection methods, and **DetectEval**, a comprehensive evaluation suite featuring five distinct out-of-distribution settings and varying intensity levels within the same LLM role.

*3.2.1 HNDC.* The HNDC consists of 16,076 human-written articles, leading to a total dataset size of 64,304 articles. For training supervised detection methods, we randomly split the HNDC into training, validation, and test sets in a 7:2:1 ratio, ensuring balanced data distribution across all sets. The test set is used to evaluate the performance of all baseline models.

*a. Human-Author News Collection.* The human-written news articles, categorized as Human-Author, are sourced from two reputable newspapers, the *New York Times* and the *Guardian*, both known for their commitment to high-quality journalism. Specifically, we extract news samples directly from the existing data

sources N24News [50] and Guardian News Articles[5], concentrating only on three domains: business, education, and technology. Each news article includes a headline and the publication date, while *New York Times* articles also include an abstract. To ensure that the articles are purely human-authored, we limit our selection to articles published before 2019, prior to the emergence of ChatGPT. In total, we collect 6,882 articles from the *New York Times* and 9,194 articles from the *Guardian.*

*b. LLM-Assisted News Generation.* To generate LLM-generated news articles, we design distinct prompts based on the three proposed roles: (1) For LLM-Creator news, the prompt includes the title, available summary, topic category, and publication date to ensure factual reliability. (2) For LLM-Polisher news, the entire original article is provided. If the article is too long, it is segmented for polishing to avoid overly shortened outputs that may result from processing lengthy articles in one go. (3) For LLM-Extender news, we retain the first three sentences or up to one-third of the original text and instruct the LLM to generate the remaining content. To ensure high-quality generation, we employ role-playing prompts, assigning the LLM the role of a journalist. This approach leverages social role assignment, which has been shown to improve LLM performance consistently [55].[6] We select LLaMa3 [16] (`Meta-Llama-3-8B-Instruct`) as the writing assistant LLM.

*3.2.2 DetectEval.* DetectEval is a comprehensive evaluation suite designed to assess the transferability and robustness of detection models, specifically focusing on *Cross-context variations* and *Multi-intensity Variations*.

*a. Cross-context variations.* Cross-context variations examine data diversity across five dimensions: content publication time, prompts for generation, source LLM, cultural differences, and content domain, resulting in five out-of-distribution settings: cross-time, cross-prompt, cross-source, cross-cultural, and cross-domain. The first four settings, similar to those in the HNDC, focus on LLM-generated content within the news domain, cross-prompt and cross-source settings are directly expanded based on the test dataset from HNDC.

**Cross-time**: HNDC's pre-2019 articles reduce LLM involvement, but data leakage is still possible. To address this, we scrape 2024 *New York Times* articles and generate LLM content for different roles using the same method as HNDC.

**Cross-prompt**: LLM-assisted writing varies by prompt, even for the same role. We design five distinct prompts per LLM-assisted news role and pair them with news articles to create a diverse test set. Prompts are listed in Appendix B.

**Cross-source**: While HNDC initially used `Llama-3-8B-Instruct`, real-world scenarios involve stronger LLMs. We supplement HNDC test data with content generated from four more powerful models: `Deepseek-v2`, `Meta-LLaMA-3-70B-Instruct`, `Claude-3.5-Sonnet`, and `GPT-4o`.

**Cross-cultural**: Considering that writing and expression styles can vary across cultural contexts even within the same domain, we constructed a cross-cultural test set using news platforms [14]

---

[5]https://www.kaggle.com/datasets/adityakharosekar2/guardian-news-articles
[6]The prompts used for HNDC construction are provided in the Appendix A.

| Feature | Human-Author | LLM-Creator | LLM-Polisher | LLM-Extender |
|---|---|---|---|---|
| **Average Word Count** | 558.98±254.31 | 377.09±61.05 | 475.83±215.63 | 511.78±107.56 |
| **Average Sentence Count** | 23.94±12.46 | 16.27±3.41 | 20.40±9.92 | 21.58±5.00 |
| **Sentiment Polarity Score** | 0.09±0.07 | 0.12±0.08 | 0.10±0.08 | 0.11±0.07 |
| **Grammatical Errors** | 16.07±11.89 | 6.58±9.74 | 10.99±11.30 | 11.02±9.79 |
| **Syntactic Diversity** | 1.52±0.50 | 1.38±0.40 | 1.40±0.42 | 1.48±0.37 |
| **Vocabulary Richness** | 0.59±0.07 | 0.52±0.06 | 0.61±0.07 | 0.51±0.06 |
| **Readability Score** | 17.26±3.11 | 18.92±2.05 | 18.63±2.37 | 18.53±2.26 |

Table 1: Feature differences between news articles. The value in the corresponding cell indicates the mean ± standard deviation.

from Germany, China, and Qatar. These countries reflect significant cultural diversity, especially in terms of language and values.

**Cross-domain**: Writing styles differ across domains. We construct a cross-domain dataset with text from thesis, story, and essay domains, sourced from CUDRT [43], MGTBench [23], and CDB [30]. Specifically, we extract relevant text types from each dataset and supplement the missing categories using our defined methodology.

*b. Multi-intensity variations.* Multi-intensity variations are introduced to address the fact that, even within the same LLM-assisted writing role, the level of LLM involvement can differ. We specifically construct test data with varying degrees of LLM involvement ratio for the LLM-extender and LLM-polisher roles.

**Variable-Length Extension**: For LLM-Extender role, we set up three truncation lengths to create variable-length extensions, allowing us to evaluate whether the detectors can identify the differing levels of content expansion. For the given text $x$, this process can be described as: $E^{(n)}(x), n \in \{Low, Medium, High\}$[7] , where $n$ denotes the truncation state of the text, and $E$ represents the process of text extension by LLMs.

**Multi-Staged Polish**: For LLM-polisher role, we apply a multi-staged polish process, iterating the text polishing up to six times to evaluate whether the detection methods can identify the increasing levels of refinement. For the given text $x$, this process can be described as: $P_m(x)$, where $m$ denotes the number of polish times, and $P$ represents the polishing process by LLMs.

*3.2.3 Linguistic Feature Comparison.* To systematically illustrate the differences in content across various LLM roles, we introduce seven linguistic feature metrics. Table 1 presents a comparison of these linguistic features across the four types of news content in HNDC. Average word and sentence count measure the number of words and sentences in a news article. Sentiment polarity score represents the emotional tone of a text, ranging from -1 to 1, with higher values indicating more positive sentiment, and lower values reflecting more negative sentiment. Grammatical errors measure the number of grammatical mistakes that occur per 1,000 words. Syntactic diversity measure the structural complexity by analyzing clause patterns. Vocabulary richness measures lexical diversity, ranging from 0 to 1, with higher values indicating greater lexical variation. Readability score measures the complexity of a text,

---

[7]We randomly retain part of an article's initial sentences and ask LLMs to complete it. **Low** refers to retaining $[3, l/3]$ sentences, **Medium** retains $[l/3, 2l/3]$, and **High** retains $[2l/3, l-3]$, where $l$ is the total number of sentences.

with higher values indicating greater reading difficulty. The detailed calculation methods for the linguistic features are provided in Appendix C.

As shown in Table 1, we observe that LLM-generated news articles are generally shorter and contain fewer sentences compared to human-written news. In contrast, LLM-polished and LLM-extended news, incorporating more human inputs, are significantly richer and more comprehensive. The various types of news exhibit trivial differences in their sentiment polarity scores. From other linguistic features, human writing shows greater lexical and syntactic variation with lower reading difficulty, whereas LLM writing is more standardized, featuring fewer grammatical errors and minimal use of informal writing styles.

## 4 EXPERIMENTS

In this section, we evaluate the performance and generalization of 10 baseline detection methods (§4.1) in our LLMDect framework across the two proposed detection tasks. Firstly We train supervised detection methods on the HNDC and evaluate their test set performance, while also reporting the zero-shot LLM detector's results (§4.2). Then we evaluate the generalization and robustness of the best-performing detection models on the HNDC across two dimensions in DetectEval: cross-context (§4.3) and multi-intensity(§4.4). Furthermore, we discuss the data leakage issue when using LLMs as zero-shot detectors (§4.5) to ensure fairness in detection outcomes. Finally, we evaluate their effectiveness by using them as feature extractors (§4.6). We report the F1 score for each LLM role and evaluate the performance of each detection method on the LLM-RR task using the weighted F1 score. We report the Mean Squared Error (MSE) and Mean Absolute Error (MAE) on the LLM-IM task.

### 4.1 Baseline Detection Methods

We consider 10 baseline detection methods. To illustrate the difficulty of the two proposed detection tasks, we first focus on decoder-only LLMs, which have demonstrated exceptional performance across a range of NLP tasks, including four recent advanced models **Mistral** [25] (Mistral-7B-Instruct-v0.3), **DeepSeek** [15] (DeepSeek-V2-Chat as of June 28, 2024), **LLaMa-3** [16] (Meta-LLaMA-3-70B-Instruct), and **GPT-4o** [35] (as of May 05, 2024). We utilize these models in a zero-shot detection pattern using specifically designed prompts, as shown in Appendix D.

Additionally, we adopt two types of supervised detection methods: feature-based classifiers and PLM-based classifiers. For feature-based methods, we adopt **Linguistic**, **Perplexity** [7], and **Rank**

| | Type | Model | LLM-RR (F1 ↑) | | | | | LLM-IM (↓) | |
|---|---|---|---|---|---|---|---|---|---|
| | | | Human | Creator | Polisher | Extender | Overall | MSE | MAE |
| Zero-shot | LLM-based | Mistral-7B | 40.01 | 0.12 | 0.12 | 0 | 10.07 | 0.4334 | 0.5678 |
| | | Deepseek-v2 | 43.58 | 21.76 | 8.56 | 0.25 | 18.54 | 0.4297 | 0.5479 |
| | | LLaMA3-70B | 44.21 | 49.31 | 1.58 | 8.95 | 26.01 | 0.2488 | 0.4353 |
| | | GPT-4o | 59.32 | 64.89 | 7.86 | 29.09 | 40.29 | 0.3079 | 0.4447 |
| Supervised | Feature-based | Linguistic | 66.16 | 80.40 | 60.60 | 71.75 | 69.75 | 0.0590 | 0.1936 |
| | | Perplexity | 60.99 | 77.90 | 59.21 | 64.06 | 65.54 | 0.0663 | 0.1984 |
| | | Rank | 61.65 | 87.37 | 61.30 | 81.17 | 72.87 | 0.0540 | 0.1841 |
| | PLM-based | RoBERTa | 99.71 | 99.93 | 99.81 | 99.78 | 99.81 | 0.0019 | 0.0222 |
| | | DeBERTa | 99.75 | 99.87 | 99.72 | 99.93 | 99.82 | 0.0027 | 0.0281 |
| | | Longformer | 99.88 | 99.94 | 99.88 | 99.94 | 99.91 | 0.0013 | 0.0168 |

**Table 2: Detection Performance of 10 Baseline Methods on the HNDC Test Set. Assuming a detector predicts an LIR of 0 for all cases in the LLM-IM task, indicating no detection capability, we can get MSE(base)=0.46 and MAE(base)=0.57.**

(GLTR) [19]. Linguistic refers to the seven linguistic features discussed in §3.2.3. Perplexity, an exponential form of entropy, assesses the model's confusion, where lower values suggest a better understanding of the text and more accurate predictions. Intuitively, and as confirmed by Gehrmann et al. [19], LLM-generated texts exhibit lower entropy since they are typically more "in-distribution". Additionally, rank feature evaluates the absolute rank of words by counting how many falls within different Top-k ranks from the LLM's predicted probability distributions. Following the classical GLTR detection method, we adopt GPT2-small [38] to extract the Perplexity and Rank features. For PLM-based methods, we choose the widely adopted models as the detectors, including **RoBERTa** [56], **DeBERTa** [22], and **Longformer** [6].

## 4.2 HNDC Performance Evaluation

Table 2 shows the detection performance of 10 baseline methods on the HNDC test set for the two tasks. The results show that supervised methods outperform the zero-shot LLM detector, with fine-tuned PLM-based models consistently achieving superior performance across both tasks. In contrast, advanced LLMs face challenges in accurately detecting content they have generated themselves. In the zero-shot LLM detector setting, Mistral and Deepseek show almost no detection capability, particularly compared to a base detector that predicts an LIR of 0 for all cases, indicating a total inability to detect LLM-generated content in the LLM-IM task. GPT-4o, while demonstrating limited ability to differentiate between human-authored and LLM-generated content, struggles to detect human-LLM collaboration. When providing detection rationales, GPT-4o tends to classify content as human-authored if it includes specific details, such as citations or data, while fluent and structured content is more likely to be identified as LLM-generated. Consequently, GPT-4o encounters significant challenges in detecting complex cases of human-LLM collaboration when relying on surface-level features alone. Feature-based models show intermediate performance, with varying detection capabilities across different LLM-generated content types. Notably, detectors using only linguistic features, without language models involved, still achieve objective results, indicating discernible linguistic differences between human-authored and LLM-assisted content, providing an

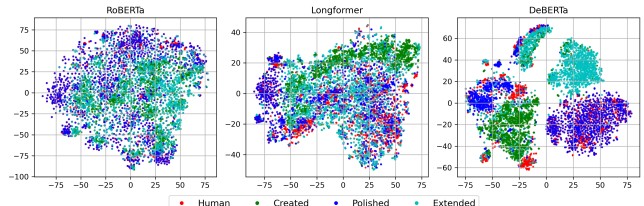

**Figure 2: t-SNE visualization of representations from three non-fine-tuned PLMs on the HNDC test data. DeBERTa shows clearer cluster separation, reflecting stronger discriminative ability.**

interpretable basis for detection. In contrast, fine-tuned PLM-based detectors exhibit outstanding detection performance across all LLM roles, indicating that traditional methods remain highly effective in detecting LLM-generated content, even in the current era of advanced LLMs.

## 4.3 Cross-context Generalization Evaluation

We apply the three best-performing PLM-based detectors trained on HNDC to five distinct types of cross-context variations in DetectEval to assess their generalization capability. Table 3 presents the generalization evaluation results. Except for the cross-domain scenario, the PLM-based detectors demonstrated strong generalization across the other four variations, achieving 90% overall. Notably, the F1 score reached 95% in the cross-time, cross-prompt, and cross-source scenarios. The high performance in these scenarios suggests that PLM-based detectors are adaptable across different contexts. Specifically, in the cross-source scenario, detectors trained on corpora generated by weaker LLMs can effectively detect content from stronger LLMs, reducing computational resources while maintaining accuracy. Interestingly, in cross-cultural settings, we find that news from countries with higher visibility, such as Germany and China, is easier to identify, while news from Qatar presents more challenges. Besides, the lower performance in the cross-domain scenario likely results from differences in language structures and

| | Test Origin | LLM-RR (F1 ↑) | | | LLM-IM (MSE ↓) | | |
|---|---|---|---|---|---|---|---|
| | | RoBERTa | DeBERTa | Longformer | RoBERTa | DeBERTa | Longformer |
| **Base** | **HNDC Test** | 99.81 | 99.84 | 99.91 | 0.0027 | 0.0013 | 0.0019 |
| **cross-time** | **Post-release** | 97.20 | 98.76 | 97.05 | 0.0150 | 0.0100 | 0.0077 |
| **cross-prompt** | **Diverse Prompts** | 95.49 | 95.04 | 96.48 | 0.0126 | 0.0119 | 0.0108 |
| **cross-source** | **LLaMa3-70B** | 99.66 | 99.81 | 99.72 | 0.0035 | 0.0019 | 0.0022 |
| | **Deepseek** | 97.96 | 98.36 | 98.94 | 0.0019 | 0.0024 | 0.0027 |
| | **Claude** | 98.18 | 98.93 | 99.02 | 0.0050 | 0.0055 | 0.0049 |
| | **GPT-4o** | 94.57 | 97.54 | 97.30 | 0.0029 | 0.0027 | 0.0040 |
| | **Average** | 97.59 | 98.66 | 98.75 | 0.0033 | 0.0031 | 0.0035 |
| **cross-cultural** | **German** | 96.22 | 94.87 | 98.18 | 0.0159 | 0.0135 | 0.0066 |
| | **China** | 97.70 | 89.24 | 98.50 | 0.0135 | 0.0079 | 0.0046 |
| | **Qatar** | 87.96 | 89.22 | 75.34 | 0.0245 | 0.0161 | 0.0291 |
| | **Average** | 93.96 | 91.11 | 90.67 | 0.0180 | 0.0125 | 0.0134 |
| **cross-domain** | **Thesis** | 68.29 | 85.88 | 81.10 | 0.0170 | 0.0179 | 0.0207 |
| | **Story** | 78.61 | 90.05 | 77.57 | 0.0357 | 0.0310 | 0.0296 |
| | **Essay** | 50.34 | 60.66 | 56.66 | 0.0598 | 0.0752 | 0.0749 |
| | **Average** | 65.75 | 78.86 | 71.78 | 0.0375 | 0.0414 | 0.0417 |
| **Overall Group Average** | | 90.00 | 92.49 | 90.95 | 0.0173 | 0.0158 | 0.0154 |

**Table 3: Generalization performance of PLM-based baseline methods across five cross-context variations.**

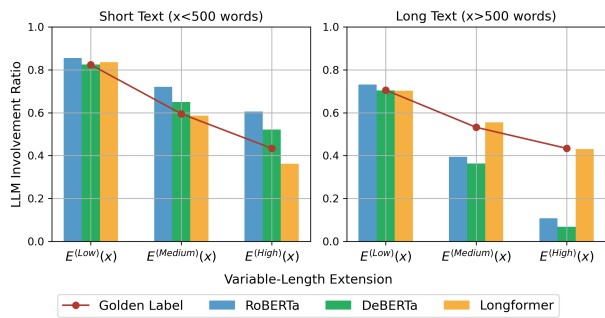

**Figure 3: Average LLM Involvement Ratio Predictions and Golden Label of Variable-Length Extension Experiments**

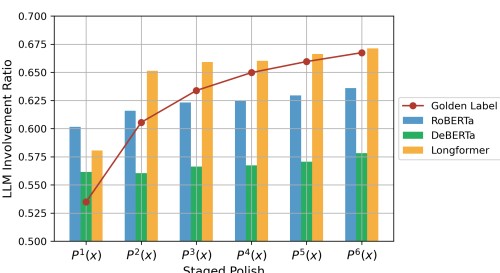

**Figure 4: Average LLM Involvement Ratio Predictions and Golden Label of Multi-Staged Polish Experiments**

terminologies, highlighting the challenge of achieving robust cross-domain adaptability and the need for domain adaptation techniques to improve detection. By averaging the generalization performance of PLM-based detectors across various cross-context groups, we find that the DeBERTa-based detector exhibit the strongest generalization capability. We hypothesize that this may be attributed to DeBERTa's use of relative position encoding, which improves its ability to capture long-range dependencies more effectively [22]. Furthermore, we input the test data from HNDC into three original, non-fine-tuned PLMs to extract their inherent learned representations, which were then visualized using t-SNE after dimensionality reduction, as shown in Figure 2. Figure 2 shows that DeBERTa achieves clearer cluster separation compared to the other PLMs, indicating stronger discriminative ability. This suggests that DeBERTa captures relevant features more effectively, explaining its superior generalization in the cross-context evaluations.

## 4.4 Multi-Intensity Robustness Evaluation

To evaluate the trained PLM-based detectors' sensitivity to LLM-generated content with different LIR levels within the same role, we apply them to the multi-intensity variations in DetectEval, assessing their robustness.[8] Figure 3 presents the results of the variable-length extension experiments.[9] Intuitively, as more original text is retained, the LLM involvement ratio decreases during text continuation. For short texts, the LIR predictions from the three PLM-based detectors generally align with the true labels, but as more original text is retained, the prediction discrepancies increase. For long texts, due to the input length limitations of RoBERTa and DeBERTa, their predicted LIR values are significantly lower than the actual values, while Longformer continues to closely match the true labels.

---

[8]Our HNDC used for training detectors only considers $E^{(Low)}(x)$ for LLM-Extender, and $P_1(x)$ for LLM-Polisher.
[9]Due to the 512-token input length limit of RoBERTa and DeBERTa, some LLM-generated texts may be truncated. Texts are categorized as long or short depending on whether they exceed 500 words.

| Architecture | Models | Params. | LLM-RR (F1 ↑) | | | | | LLM-IM (↓) | |
|---|---|---|---|---|---|---|---|---|---|
| | | | Human | Creator | Polisher | Extender | Overall | MSE | MAE |
| **Encoder-only** | **RoBERTa** | 125M | 47.41 | 68.00 | 41.74 | 51.87 | 52.25 | 0.1101 | 0.2784 |
| | **DeBERTa** | 140M | 47.55 | 57.67 | 52.85 | 63.05 | 55.28 | 0.1291 | 0.3154 |
| | **Longformer** | 149M | 46.67 | 62.47 | 36.94 | 50.12 | 49.05 | 0.1126 | 0.2815 |
| **Decoder-only** | **GPT2-small** | 117M | 61.65 | 87.37 | 61.30 | 81.17 | 72.87 | 0.0540 | 0.1841 |
| | **GPT2-medium** | 345M | 72.79 | 91.92 | 73.97 | 82.71 | 80.35 | 0.0546 | 0.1858 |
| | **GPT2-large** | 774M | 72.82 | 92.25 | 74.38 | 83.06 | 80.63 | 0.0544 | 0.1853 |
| | **Mistral-7b** | 7B | 59.00 | 95.29 | 54.40 | 52.79 | 65.37 | 0.0682 | 0.2040 |
| | **LLaMa3-8b** | 8B | 65.09 | 94.33 | 63.01 | 86.51 | 77.24 | 0.0570 | 0.1847 |

Table 4: Features-based Detectors From Different Language Models

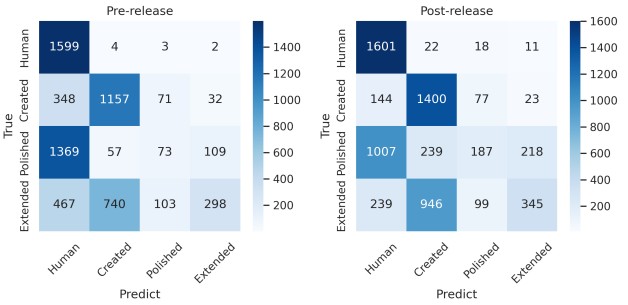

Figure 5: Comparison of Confusion Matrices of LLM-RR Task

Figure 4 shows the results of the multi-staged polish experiments. As the number of polishing stages increases, the LIR value rises accordingly. Longformer provides the best fit for predicting LIR compared to the other two models. This is because accurately estimating the LIR of human-LLM collaborative content requires a comprehensive evaluation of the entire input text. Longformer's ability to handle longer inputs allows it to extract more complete features, making its LIR estimates more robust.

### 4.5 Data Leakage analysis

A key concern when using LLMs as detectors to distinguish human-author and LLM-generated content is that the LLMs' training corpus may include these news data sources. To address this issue and ensure fairness in evaluation, we select the HNDC test set as a pre-release dataset and cross-time data from DetectEval as a post-release dataset[10], using the best-performing zero-shot LLM, GPT-4o, to conduct LLM-RR experiments on both datasets. As shown in Figure 5, which presents confusion matrices for GPT-4o's performance on pre-release and post-release data, we find that data leakage issue actually reduces performance. In the pre-release dataset, almost all LLM-Polished news articles are misclassified as human-authored. In the post-release dataset, the proportion decreases, likely due to GPT-4o's prior exposure to human-authored news, proving that data leakage affects LLM judgment. Furthermore, as shown in the confusion matrices, the proportion of misclassifications in the post-release dataset has decreased. This suggests that exposure to news

---

[10]GPT-4o has a knowledge cutoff date of October 2023, and since the selected news data comes from 2024, it helps to avoid the data leakage issue.

during training misleads the judgment of zero-shot detectors, which could also explain the poor performance reported in the literature [8] when using LLMs to distinguish LLM-generated news from human-authored news.

### 4.6 LLMs Feature Extractors Analysis

Although generative decoder-only LLMs perform poorly in zero-shot detection, fine-tuning these models is computationally expensive, and PLM-based detection models have already achieved outstanding performance. Nevertheless, we can explore the potential of using these LLMs directly as feature extractors. Specifically, drawing on the GLTR [19] approach, we train detection models using rank-based features extracted from different LLMs. The detection performance of each model when used as a feature extractor is presented in the Figure 4. We observe that decoder-only models, such as GPT-2, significantly outperform encoder-only models like RoBERTa, DeBERTa, and Longformer, even with comparable parameters[11]. Among categories of LLM-assisted writings, LLM-creator texts are easiest to distinguish, whereas distinguishing between LLM-Polisher and Human-Author texts remains challenging. Additionally, comparisons among different size of GPT-2 reveal that larger models demonstrate better feature effectiveness.

### 5 Conclusion

In this paper, we introduce a new detection paradigm that moves beyond binary classification by considering both the role and level of LLM involvement in content creation. We proposed two novel tasks: LLM Role Recognition (LLM-RR) and LLM Influence Measurement (LLM-IM), offering a more fine-grained approach to detecting LLM-generated content. To support these tasks, we develop LLMDetect, a benchmark combining the Hybrid News Detection Corpus (HNDC) and DetectEval, designed to assess model robustness and generalization across diverse contexts. Our empirical evaluation of 10 baseline detection models demonstrated that fine-tuned PLM-based methods outperform others, with DeBERTa excelling in cross-context generalization and Longformer performing best with varying intensity levels. As LLM-generated content becomes more prevalent, particularly on social media, these findings highlight the importance of developing more effective and fine-grained detection models. Our approach provides valuable tools for detecting LLM involvement, contributing to improved content integrity in digital platforms.

---

[11]https://huggingface.co/transformers/v4.11.3/pretrained_models.html

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

# A  Prompts for HNDC Construction

The designed role-playing prompts for HNDC construction are shown in Table 5.

# B  Diverse Prompts

We assign five prompts to each role, with each news in the test set paired with a single prompt and call LLaMA-3 to generate the cross-prompt dataset. The prompts are shown in Table 6.

# C  Supplement on Linguistic Features

The detailed calculation methods of linguistic features are as follows: (1) **Sentiment polarity score**: We use the VADER sentiment analysis package to calculate sentiment polarity score. (2) **Grammatical errors**: We use the LanguageTool package to check for grammatical errors. (3) **Syntactic diversity**: Specifically, it is measured by calculating the ratio of the number of subordinate clauses to the total number of sentences. We use the spacy package to segment clauses. (4) **Vocabulary richness**: We assess lexical diversity using the Type-Token Ratio (TTR), which is the ratio of unique tokens to total tokens in a text. (5) **Readability score**: We use Fog Index to assess readability, which indicates the number of years of education required to understand the text. A higher Fog Index value represents lower readability, calculated using the average sentence length and the percentage of words with three or more syllables.

# D  LLM Detectors Prompt

Table 7 presents the instruction prompts used by the zero-shot LLM detector for the two detection tasks.

# E  GLTR Visualization

Figure 6 shows a visualization of the absolute ranks of words in news articles generated by different methods. Human-authored news, as shown in (a), contains a large number of red or purple words, indicating low ranks. In contrast, LLM-created news, as shown in (b), features few red or purple words, with most words marked in green or yellow, indicating higher ranks. LLM-polished news, depicted in (c), shows a significant decrease in the proportion of high-rank words. Meanwhile, LLM-extended news, illustrated in (d), shows that the initial human-authored part contains many low-rank words, while the subsequent LLM-generated continuation predominantly uses high-rank words. This partly explains the differences in the texts generated through the different roles of LLMs in content creation.

| **LLM-Creator** | |
| --- | --- |
| **System Prompt:** | You are an AI assistant tasked with generating news articles. **Given a news article title and its description, your task is to craft a well-structured and informative news article.** Aim for a balanced and informative article that provides context and clarity to the reader. Adapt the tone and style to fit the nature of the news, whether it's business, education, or scientific to engage the target audience effectively. |
| **User Prompt:** | Here is a news article title: <title> and its description <description>, write a <category> news article based on this news article title and description I gave you and return news article as well as your title with the format Title: __ ### Article: __ (make sure to use ### as the delimiter). The article should reflect information available up to <publish date>. |
| **LLM-Polisher** | |
| **Prompt:** | Please rewrite the following sentences.
"'<news articles>'" |
| **LLM-Extender** | |
| **System Prompt:** | You are an AI assistant tasked with generating news articles. **Your task is to continue writing from the given incomplete news article and ensure the continuation is well-structured and informative.** Aim for a balanced and informative article that provides context and clarity to the reader. Adapt the tone and style to fit the nature of the news, whether it's business, education, or scientific to engage the target audience effectively. |
| **User Prompt:** | Please complete the following news article. Don't return the given text.
The news begin with:
"'<beginning text>'"
Continue from here: |

**Table 5: The Designed Prompts for HNDC Construction**

| **LLM-Creator** | |
| --- | --- |
| **Prompt1:** | Here is a news article title: \<title\> and its description \<description\>, write a \<category\> news article based on this news article title and description I gave you and return news article as well as your title with the format Title: \_\_ ### Article: \_\_ (make sure to use ### as the delimiter). The article should reflect information available up to \<publish date\>. |
| **Prompt2:** | Here is a news article title: \<title\> and its description \<description\>, write a \<category\> news article based on this news article title and description I gave you and return news article as well as your title with the format Title: \_\_ ### Article: \_\_ (make sure to use ### as the delimiter). |
| **Prompt3:** | Please write a news about \<title\>, \<description\>. Return news article as well as your title with the format Title: \_\_ ### Article: \_\_ (make sure to use ### as the delimiter). |
| **Prompt4:** | Here is a news article title: \<title\> and its description \<description\>, write a news article in an expert confident voice. Return news article as well as your title with the format Title: \_\_ ### Article: \_\_ (make sure to use ### as the delimiter). |
| **Prompt5:** | Please write a news about \<title\>, \<description\> in a formal scientific writing voice. Return news article as well as your title with the format Title: \_\_ ### Article: \_\_ (make sure to use ### as the delimiter). |
| **LLM-Polisher** | |
| **Prompt1:** | Please polish the following sentences.
"'\<news article text\>'" |
| **Prompt2:** | Please enhance fluency of the following sentences.
"'\<news article text\>'" |
| **Prompt3:** | Please adjust structures of the following sentences.
"'\<news article text\>'" |
| **Prompt4:** | Please rewrite the following sentences in a formal scientific writing voice.
"'\<news article text\>'" |
| **Prompt5:** | Please polish the following sentences in a humorous voice.
"'\<news article text\>'" |
| **LLM-Extender** | |
| **Prompt1:** | Please complete the following news article. Don't return the given text.
The news begins with:
"'\<beginning text\>'"
Continue from here. |
| **Prompt2:** | Please directly continue to write the news (not repeat my provided content):
"'\<beginning text\>'" |
| **Prompt3:** | Please complete the following news article. Don't return the given text.
The text begin with:
"'\<beginning text\>'" |
| **Prompt4:** | Complete the following unfinished news article. Don't return the given text.
The news begin with:
"'\<beginning text\>'"
Continue from here. |
| **Prompt5:** | Please directly continue to write the news (not repeat my provided content):
The news begin with:
"'\<beginning text\>'"
Continue from here. |

**Table 6: The Diverse Prompts for Cross-prompt of DetectEval**

**LLM-RR Prompt**

Your task is to identify the generated method of the provided <Article>.
The candidate options include:
A. Human-Written: The article is written entirely by humans without any AI assistance;
B. AI-Created: The article is generated by AI entirely from a given topic;
C. AI-Polished: The article is polished by AI from a human-written draft;
D. AI-Extended: The article is initially written by humans, and then additional content is generated by AI to expand on the original material.

Please directly give the answer with answer-rationale pair in JSON format, with the structure:
"answer": ..., "rationale": ...."
Please directly give the "answer" with "A", "B", "C", or "D", and explain your choice in two or three sentences (string format) in "rationale".

**LLM-IM Prompt**

Your task is to evaluate the extent of AI-assisted writing in the provided article.

The evaluation scores range from 0 to 1, where 0 indicates the article is completely human-written, and 1 indicates it is entirely AI-created.
Please directly give the answer with score in JSON format, with the structure: "score": ...

**Table 7: The Prompts for LLMs as Zero-shot Detectors**

(a) Human-Author

(b) LLM-Creator

(c) LLM-Polisher

(d) LLM-Extended

**Figure 6: GLTR visualization results of sample texts. A word that ranks within the top 10 probability is highlighted in green, top 100 in yellow, top 1,000 in red, and the rest in purple.**

