# OpenReview forum: "Beyond Binary: Towards Fine-Grained LLM-Generated Text Detection via Role Recognition and Involvement Measurement"
_ACM.org/TheWebConf/2025/Conference — WWW 2025 Poster_

### Official Review · Reviewer_RQLA · 2024-11-29

**Novelty:** 6
**Technical Quality:** 6

**Review:**

The paper introduces a novel approach to detecting LLM-generated content through multi-class classification and regression tasks. These contributions aim to address the constraints of binary classification and improve generalizability across diverse contexts. While the study is a promising step forward in content detection methodologies, a number of limitations should be addressed.

Pros:
 - The paper addresses a pertinent topic, advancing detection methods for LLM-generated content
 - The authors plan to share the datasets and evaluation frameworks with the broader community
 - The work proposes a novel extension to the binary classification of LLM-assisted content

Cons:
 - The training dataset is limited in scope, relying on articles from specific news domains and a single LLM
 - Prompts are reused between the training and evaluation datasets
 - The paper provides insufficient detail about DetectEval to properly evaluate the results

Major Comments:
 - CCS Concepts: The paper's CCS Concepts should be generated appropriately.
 - Dataset Construction and Generalizability:
   - The Hybrid News Detection Corpus (HNDC) heavily relies on articles from The New York Times and The Guardian in business, education, and technology domains. This domain specificity likely reduces the generalizability of results to other contexts.
   - This dataset is generated using a single LLM (LLaMa-3-8b) and five prompt variations for each task type, limiting training data variety.
   - While the DetectEval benchmark expands the variability by including data generated by other models, it is hard to evaluate this improvement given the lack of summary statistics and in-depth descriptions.
   - DetectEval is further used for benchmarks, making the more detailed description essential for the evaluation the out-of-distribution performance.
 - Methodology and Metrics:
   - It needs to be clarified why the authors decided on a 7:2:1 train-validation-test split instead of cross-validation.
   - The reliance on zero-shot LLMs for evaluation while training fine-tuned PLM-based models like DeBERTa skews the results in favor of the fine-tuned models. Including fine-tuned LLMs in the comparison would provide a more balanced assessment.
   - Table 1 and Section 3.2.3 primarily focus on HNDC with LLaMa-3-8b. Considering that the main emphasis of the paper seems to be on the DetectEval framework and its use for testing generalizability across multiple dimensions,  it would be more insightful to see differences across other LLMs on a more diversified DetectEval data, as well as the summary statistics of the data within DetectEval.
 - LLM Role Recognition and LLM Influence Measurement:
   - The necessity of the LLM-RR task needs to be clarified further. LLM-IM regression values already indicate the extent of LLM involvement (from 0 for human-authored to 1 for LLM-generated), potentially making the classification into four roles (Human-Author, LLM-Creator, LLM-Extender, LLM-Polisher) redundant. The authors should clarify whether LLM-RR provides additional practical benefits or improved detection performance.
   - The practical distinction between LLMDect and LLM-RR might be clearer.
 - Overfitting and Prompt Reuse:
   - Reusing prompts for both HNDC (training data) and DetectEval (evaluation framework) raises concerns about overfitting, as models may adapt to these specific prompts rather than generalizing to unseen scenarios. This is supported by the model's degraded performance in cross-domain evaluations, where essay-type detection showed F1 scores lowered by up to half and MSE increased by an order of magnitude, suggesting that models may overfit the news domain and specific writing styles seen during training.
   - The study focuses heavily on news articles. While extensions into other domains are mentioned as part of DetectEval, they are lacking in-depth descriptions.
 - Evaluation Benchmarks and Perplexity Features:
   - The evaluation relies on GPT-2 small for perplexity and rank features. Authors should comment on the choice of this model.

Minor Comments:
 - The finding that "advanced LLMs face challenges detecting their own content" is unsurprising, given the comparison of fine-tuned PLM-based models with zero-shot LLMs.
 - Using only news articles with specific stylistic patterns may further reduce transferability to other domains.
 - Table 4 is not referenced in text
 - On line 387 the authors switch from LLMDect to LLMDetect, which is also mentioned in the abstract. Are they the same thing?

Typos:
 - [Line 19] roles of[]LLM
 - [Line 515] syntactic diversity measure[s]
 - [Line 693] present[] more.
 - [Line 694] still detect objective[ly reasonable] results. It might not be a typo, but objective results sound incomplete
 - In table 3, it should probably say German[y]

This paper presents a promising framework for detecting LLM-generated content using novel multi-class classification and regression tasks. However, limitations in dataset diversity, over-reliance on prompts from a single LLM, and insufficient details about evaluation benchmarks undermine its results. Nevertheless, the contributions and availability of the data and methods make this paper a valuable contribution.

**Questions:**

1. Could the authors comment on the potential over-fitting caused by reusing prompts from HNDC in DetectEval?
2. Given that LLM-IM already provides a continuous measure of LLM involvement (e.g., from 0 to 1), does LLM-RR provide additional practical benefits or improved detection accuracy?
3. Why does the evaluation rely on GPT-2 small for perplexity and rank features?
4. Could the authors provide more details and summary statistics for DetectEval? For example, how many tasks and domains are included, and what percentage of the data comes from extensions beyond the news domain.
5. Why was the 7:2:1 train-validation-test split chosen instead of cross-validation?
6. Could the authors expand Table 1 to include the data from DetectEval?

**Reviewer Confidence:**

3: The reviewer is confident but not certain that the evaluation is correct

**Scope:**

4: The work is relevant to the Web and to the track, and is of broad interest to the community

---

### Official Review · Reviewer_PAY3 · 2024-11-30

**Novelty:** 5
**Technical Quality:** 4

**Review:**

This paper presents a new paradigm for detecting LLM generated content consisting of two novel tasks, namely LLM Role Recognition and LLM Influence Measurement. The former is a multi-class classification task aimed at identifying the specific role a LLM had in content generation (e.g., completion, rephrasing, etc.), whereas the latter refers to the degree of involvement the LLM had in content creation. Besides this, the authors propose a new benchmark (LLMDetect) and training corpus (Hybrid News Detection Corpus). The proposed approach is tested using 10 baseline detection methods encompassing both encoder and decoder models.

*Strengths*
- As human-AI interactions are increasingly common, moving beyond binary (i.e., human or machine) LLM-generated text detection is paramount, and this paper offers an interesting perspective on the human-machine combinations.
- The development of the LLMDetect benchmark provides further tools to advance our detection capabilities.
- Some of the considered baseline detection models demonstrate remarkable capabilities on test data considering specific settings.

*Weaknesses*
- The boundaries between some roles (e.g., polisher vs. extender) seem a bit weak to me, as editing/refining can relate to extending text, and vice-versa. While this boundary can be more distinguishable in the case of low/medium retaining, things can become more noisy in the high retaining scenario.
- While using the Jaccard distance for computing the polish ratio can be effective as already proven in the literature, I wonder what happens when an LLM polishes a given text by simply rearranging sentences and/or keeping most of the words as they are.
- There are no technical details on the generation/assistance steps performed via LLM, such as temperature, top_p and top_k, and decoding strategy. These are paramount for reproducibility as well as for understanding how the generation was carried out. Similar considerations hold for the fine-tuning hyperparams.
- The results reported in Table 3 suggest limited generalizability to other domains, with remarkable drops in F1.
- Related to previous point, while focusing on news articles is a common choice due to the presence of topical and structural nuances, I wonder if the reported findings generalize to shorter sentences (e.g., Tweets), that might be harder to discern.

**Questions:**

- Are the reported F1 scores the result of cross-validation and/or from multiple runs, or just deriving from a single run?
- I wonder why the authors use just one relatively small LLM as the writing assistant (i.e., Llama3 3B). What can be the implications on this choice on the results? That is, does using larger models or multiple models have an impact on the detectability?
- As a suggestion more than a question, some more details on interpretability/explainability as well as error analysis based on the reported findings would be appreciated and might help steer future directions on AI-generated text detection.
- It would be helpful to complement Figure 2 with a fine-tuned version, so as to highlight the difference in the latent space, given the remarkable detection capabilities reported for these PLMs.
- In Section 4.2 the authors state that LLMs struggle in detecting "themselves", however this seems to contrast with existing literature [1]. Is this due to the fact that models like Mistral and Deepseek have to detect Llama3-generated text? In this case, the sentence should be revised to make it more clear.

[1] "LLM Evaluators Recognize and Favor Their Own Generations". Panickssery et al., NeurIPS2024.

**Reviewer Confidence:**

3: The reviewer is confident but not certain that the evaluation is correct

**Scope:**

3: The work is somewhat relevant to the Web and to the track, and is of narrow interest to a sub-community

---

### Official Review · Reviewer_h9C1 · 2024-12-01

**Novelty:** 4
**Technical Quality:** 5

**Review:**

Overview:

This paper proposes a new paradigm for detecting LLM-generated content by moving beyond simple binary classification. It introduces two novel tasks: LLM Role Recognition (LLM-RR) for identifying specific roles of LLMs in content generation, and LLM Influence Measurement (LLM-IM) for quantifying the extent of LLM involvement. The authors also present LLMDetect, a benchmark consisting of the Hybrid News Detection Corpus (HNDC) and DetectEval evaluation suite, designed to assess model robustness across diverse contexts.

Pros:

1.	Clear and well-structured presentation with thorough problem definition, detailed methodology description, and comprehensive analysis of results

2.	Comprehensive experimental design that evaluates multiple aspects of LLM-generated content detection, testing baseline models across various dimensions using their proposed datasets and metrics

Cons:

1.	Some of the paper's claimed contributions are rather straightforward and lack significant novelty. Specifically, the extension from binary classification to a four-category classification system and the development of a regression task for measuring LLM influence level are not original contributions, as some public AI text detection tools released in the past two years can already provide percentage estimates of AI involvement in texts.

2.	While the corpus sources (New York Times and The Guardian) ensure high-quality content, the dataset is limited to only news-style writing, failing to include other important text types such as personal correspondence between individuals, academic papers by researchers, and other forms of written communication encountered in real-life scenarios.

**Questions:**

1. Why is there a need to separately define LLM-RR and LLM-IM tasks? It seems that LLM-RR could be subsumed under LLM-IM, with LLM-RR being a coarse-grained version and LLM-IM being a fine-grained version of essentially the same task.

2. Are the four categories in LLM-RR truly comprehensive? In real-world scenarios, there can be hybrid cases combining LLM-Polisher and LLM-Extender roles. For example, when I provide the main content of an article and ask an LLM to both polish the main themes and generate a summary of the article.

3. In your description of DetectEval, you mention supplementing the HNDC dataset with stronger LLMs like Claude-3.5-Sonnet and GPT-4o in the cross-source section. What is the extent of this supplementation? Did you regenerate the entire HNDC dataset (originally generated by Meta-Llama-3-8B-Instruct) using GPT-4o and other LLMs, or did you only generate additional partial data?

4. In Table 1, the sentence length of LLM-Creator-generated text is notably shorter than others. Could sentence length be directly used as a specific feature by subsequent baseline models to identify whether a text is LLM-Creator generated?

5. Among the baseline models, the authors only used zero-shot LLMs for text detection. Did the authors explore other methods to enhance LLM performance, such as few-shots or RAG, to test the potential of LLMs in text detection?

**Reviewer Confidence:**

3: The reviewer is confident but not certain that the evaluation is correct

**Scope:**

3: The work is somewhat relevant to the Web and to the track, and is of narrow interest to a sub-community

---

### Official Review · Reviewer_bcB8 · 2024-12-02

**Novelty:** 5
**Technical Quality:** 5

**Review:**

This paper addresses the challenge of detecting text generated or assisted by large language models (LLMs), such as ChatGPT. Instead of focusing on simple binary classification (human vs. AI-generated), the authors propose two new tasks: identifying the specific roles LLMs play in content creation (e.g., drafting, editing, or extending) and measuring the degree of LLM involvement in the text. They introduce a benchmark called LLMDetect, which includes a diverse dataset and evaluation framework to test model robustness across real-world contexts. Their experiments show that fine-tuned models like DeBERTa outperform others in detecting LLM-generated content, while advanced LLMs struggle to detect their own outputs.


Strengths:

Overall, this paper addresses novel research questions, particularly with a more realistic setup for detecting LLM-generated text, where an LLM only partially generates the text, simulating a more human-LLM collaboration.

The experiments are exhaustive, exploring many generalization setups.

Weaknesses:

The regression task predicts the ratio of LLM-generated text compared to the total text or the Jaccard distance for the LLM-polisher scenario. However, these feel like completely different things, and having a single regression model to predict both needs more discussion. One issue is that Jaccard's "distance" will provide a distance that converges to zero as the number of changes using an LLM increases, while the original ratio from Equation 3 will converge to 1. Even if the Jaccard index was used instead of Jaccard similarity, the two constructs (ratio vs similarity) are measuring different things. Maybe using Jaccard similarity for everything would be more consistent.

The new benchmark is mentioned as a major contribution of the paper. However, based on the results, many of the results are over .9 F1 for detection, even in the cross-* setups (e.g., cross-source). This suggests a limited opportunity for future methodological advancement using this benchmark. One suggestion is to explore a more intersectional aspect to evaluation. Yes, this increases experiments. But, it would be interesting to see if testing on both out-of-domain LLMs across out-of-domain cultural aspects has a much larger impact on performance. Intersectional aspects are more likely in practice and could make the benchmark more usable.

Finally, it is unclear how to understand generalization in this paper with regard to source model type compared to prior work (i.e., how to understand performance on unseen models). On the one hand, yes, the experiments seem to measure generalization okay. However, prior work [1] seems to have much more variation when testing across different model types. But maybe newer models have converged to have similar stylistic patterns? I feel like there needs to be more analysis and discussion on this point. I feel that showing many .9 scores for detection can be harmful when generalizations may not be like that in practice. Maybe a notification that practical performance can vary more would suffice.

References:

[1] Wang, Yuxia, et al. "M4: Multi-generator, Multi-domain, and Multi-lingual Black-Box Machine-Generated Text Detection." Proceedings of the 18th Conference of the European Chapter of the Association for Computational Linguistics (Volume 1: Long Papers). 2024.

**Questions:**

1. Can you provide some reasoning for why Jaccard distance can be treated the same as Equation 3?
2. How should researchers use the benchmark? Should the focus be on the categories that have room for improvement (e.g., cross-domain)? What is the use for the other aspects (e.g., cross-time)?
3. What do you think is the generalization across model types in practice?

**Reviewer Confidence:**

3: The reviewer is confident but not certain that the evaluation is correct

**Scope:**

4: The work is relevant to the Web and to the track, and is of broad interest to the community

---

### Official Review · Reviewer_Cc53 · 2024-12-03

**Novelty:** 7
**Technical Quality:** 7

**Review:**

I enjoyed reading this manuscript. I think the work is important and well-executed.  The manuscript is clear.

pros:
- important problem
- clear manuscript
- good idea to split the problem into role recognition and level of involvement.
- very thorough benchmark
- results look promising

cons:
- no real issues.

minor comments:
- There is an inconsistency in the term "LLM *Influence* Measurement (LLM-IM)" vs "LLM *Involvement* Measurement (LLM-IM)"
- in eq. 1, the data distribution D should be introduced (it's done in eq. 4).
- "Therefore, following the approach of Yang et al. [53], we use the Jaccard distance to calculate the polish ratio,": could you clarify how it is computed? based on the set of words?
- Figures legends are too small.
- " The detection performance of each model when used as a feature extractoris presented in the Figure 4." should be Table 4.

**Questions:**

No questions.

**Reviewer Confidence:**

3: The reviewer is confident but not certain that the evaluation is correct

**Scope:**

4: The work is relevant to the Web and to the track, and is of broad interest to the community